# Inhibition of Human Cytomegalovirus Entry into Host Cells through A Pleiotropic Small Molecule

**DOI:** 10.3390/ijms21051676

**Published:** 2020-02-29

**Authors:** James Elste, Dominik Kaltenbach, Vraj R. Patel, Max T. Nguyen, Harsh Sharthiya, Ritesh Tandon, Satish K. Mehta, Michael V. Volin, Michele Fornaro, Vaibhav Tiwari, Umesh R. Desai

**Affiliations:** 1Department of Microbiology & Immunology, College of Graduate Studies and Chicago College of Osteopathic Medicine, Midwestern University, Downers Grove, IL 60515, USA; jelste@midwestern.edu (J.E.); vpatel23@midwestern.edu (V.R.P.); mnguyen98@midwestern.edu (M.T.N.); MVOLIN@midwestern.edu (M.V.V.); 2Department of Biomedical Sciences, College of Graduate Studies, Midwestern University, Downers Grove, IL 60515, USA; dkaltenbach29@midwestern.edu; 3Department of Anatomy, College of Graduate Studies and Chicago College of Osteopathic Medicine, Midwestern University, Downers Grove, IL 60515, USA; harshsharthiya@gmail.com (H.S.); mforna@midwestern.edu (M.F.); 4Department of Microbiology and Immunology, University of Mississippi Medical Center, 2500 North State Street, Jackson, MS 39216, USA; rtandon@umc.edu; 5KBR Wyle Laboratories, Houston, TX 77058, USA; satish.k.mehta@nasa.gov; 6Department of Medicinal Chemistry, School of Pharmacy, Virginia Commonwealth University, Richmond, VA 23298, USA; 7Institute for Structural Biology, Drug Discovery and Development, Virginia Commonwealth University, Richmond, VA 23219, USA

**Keywords:** human cytomegalovirus infection, virial entry, virus-cell interaction, herpesviruses, heparan sulfate, heparan sulfate mimetics

## Abstract

Human cytomegalovirus (HCMV) infections are wide-spread among the general population with manifestations ranging from asymptomatic to severe developmental disabilities in newborns and life-threatening illnesses in individuals with a compromised immune system. Nearly all current drugs suffer from one or more limitations, which emphasizes the critical need to develop new approaches and new molecules. We reasoned that a ‘poly-pharmacy’ approach relying on simultaneous binding to multiple receptors involved in HCMV entry into host cells could pave the way to a more effective therapeutic outcome. This work presents the study of a synthetic, small molecule displaying pleiotropicity of interactions as a competitive antagonist of viral or cell surface receptors including heparan sulfate proteoglycans and heparan sulfate-binding proteins, which play important roles in HCMV entry and spread. Sulfated pentagalloylglucoside (SPGG), a functional mimetic of heparan sulfate, inhibits HCMV entry into human foreskin fibroblasts and neuroepithelioma cells with high potency. At the same time, SPGG exhibits no toxicity at levels as high as 50-fold more than its inhibition potency. Interestingly, cell-ELISA assays showed downregulation in HCMV immediate-early gene 1 and 2 (IE 1&2) expression in presence of SPGG further supporting inhibition of viral entry. Finally, HCMV foci were observed to decrease significantly in the presence of SPGG suggesting impact on viral spread too. Overall, this work offers the first evidence that pleiotropicity, such as demonstrated by SPGG, may offer a new poly-therapeutic approach toward effective inhibition of HCMV.

## 1. Introduction

Human cytomegalovirus (HCMV), also known as human herpesvirus 5, is a widespread pathogen among human populations [1,2,3]. To establish a successful infection, HCMV depends on a unique set of immune evasion strategies, using encoded proteins and mRNA to avoid a competent innate and adaptive immune response [4]. HCMV-mediated infections are a recognized cause of morbidity and mortality in immunocompromised individuals as compared to otherwise healthy or immunocompetent individuals [5,6,7]. During the early stages of fetus development, HCMV infection in utero can result in disseminated infection with severe neurological impairments [8,9,10,11,12,13,14]. In addition, HCMV infection can also cause visual impairments in the form of retinitis, which is commonly seen in the AIDS population [15,16,17]. Interestingly, HCMV is capable of infecting multiple cell and tissue types, which suggests its ability to interact with a diverse array of host cell receptors to establish an infection [18,19,20].

HCMV entry into host cells is extremely dynamic due to the usage of a variety of cellular entry receptors during virus-cell interactions [21,22,23,24]. It has been shown that depending on the cell type, HCMV can utilize multiple target cell receptors, including cell surface heparan sulfate (HS), different cellular integrins, platelet-derived growth factor receptor alpha (PDGF-Rα), neuropilin-2 and epidermal growth factor receptor (EGF-R), for cellular entry [25,26,27,28,29,30,31]. On the flip side, HCMV’s multiple viral glycoproteins, including gB, gO, and gH-gL, engage these structurally and functionally unrelated receptors to penetrate cellular defenses. For example, gB catalyzes membrane fusion between HCMV and infected cells, while gH/gL-containing complexes regulate viral tropism [24]. Interestingly, these viral glycoproteins contribute differently to different cell-types. Whereas the trimer of HCMV’s gH-gL-gO is critical for infection in fibroblasts, a pentamer unit of gHgLpUL128-131A is essential for viral entry in epithelial, endothelial, and myeloid cells [32,33,34]. The distinctive ability of HCMV to utilize a diversity of cell surface receptors coupled to its exploitation of multiple glycoproteins to mediate entry pose a special challenge for drug discovery because of the possibility of several bypass mechanisms.

Few drugs are currently available for HCMV treatment including ganciclovir, valganciclovir, foscarnet, cidofovir, and fomivirsen [35]. Yet, as a group these carry a number of limitations including toxicity arising from prolonged therapy, limited effectiveness, poor bioavailability, and acquired resistance to common targets [35,36,37,38,39]. The discovery of new anti-HCMV agents with better efficacy and novel mechanism of action is a critical goal [40]. More specifically, for HCMV it would beneficial to develop an agent that could simultaneously target multiple receptors involved in viral entry so as to more effectively prevent bypass pathways.

One group of receptors that could be targeted by such agents include the heparan sulfate proteoglycans. First, HCMV glycoproteins, especially gB, are known to bind to HS chains present on these cell surface proteoglycans [41]. Thus, appropriate competitive molecules would be expected to reduce cellular attachment and entry. Second, many host cell surface receptors are also known to bind to HS such as growth factor receptors (e.g., PDGF-R and EGF-R) and neuropilin [28], which could offer additional opportunities for competitive inhibition. Third, inhibitors of viral entry could be expected to have downstream consequences such as the expression of viral immediate-early genes, whose products play a key role in the pathogenesis of HCMV infection, and prevention or reduction in host immunomodulation [42].

In fact, the literature is replete with the use of heparin, heparan sulfate, and other sulfated polysaccharides as anti-viral agents [43,44,45,46]. This includes targeting HSV [47], HIV [48,49], and others. Yet, a problem with the use of these biopolymers is that it is extremely difficult to synthesize and characterize them. It is also very difficult to transform the polysaccharide scaffold into potential drug-like scaffold, which would help further drug design and development efforts.

For some time now, we have been pursuing non-polymeric, non-saccharide mimetics of heparin/heparan sulfate as anti-viral [50,51,52,53,54], anti-cancer [55,56], and anti-thrombotic agents [57,58]. The advantage with these mimetics is that these are fully synthetic, easily characterizable, and highly amenable to advanced rational drug design possibilities. We now have a library of some 150 non-saccharide mimetics of heparin/heparan sulfate with different structural scaffolds, diversity of sulfate groups, and molecular sizes [59,60]. One of these agents is sulfated pentagalloylglucoside (SPGG) with a distinct structure to simultaneously mimic many different sequences of heparin/heparan sulfate (Figure 1). The unique aspect of SPGG is its globular structure that theoretically can interact with diverse HS-binding proteins. In fact, our earlier work on SPGG has shown that it interacts with coagulation factor XIa [57,61] and glycoprotein gD [50], two completely different HS-binding proteins. Thus, we reasoned that the probability of targeting multiple proteins involved in cellular entry of HCMV, e.g., glycoprotein gB, PDGF-R, EGF-R, and neuropilin, would be higher with SPGG than with other non-saccharide mimetics developed so far.

Although pleiotropicity, i.e., ability to bind to multiple targets, is dis-favored for drug discovery and development, the case of HCMV is unique. HCMV relies on multiple targets to gain access to different cell types and inhibiting all these pathways with a ‘magic’ bullet is likely to be extremely challenging. At the same time, without the development of such a pleiotropic inhibitor, the virus would probably find escape routes resulting in sub-optimal activity. Thus, we reasoned that SPGG is likely to afford the highest chance of success against HCMV, which could then become useful springboard for further development.

## 2. Results

### 2.1. Effect of SPGG Treatment on Cell Survival

To first assess whether SPGG is toxic to cells typically targeted by HCMV, we studied dose-dependence of cell viability. We utilized lactate dehydrogenase (LDH) assay as a surrogate for cell viability, as described in the literature [62]. Briefly, rupture of plasma membranes due to apoptosis, necrosis, and other forms of cellular damage results in release of LDH into the cell culture supernatant, which can be quantified by an NADH coupled chromogenic reaction. We selected two different cell lines, Human Foreskin Fibroblasts (HFF-1) and neuroepithelioma cells (SK-N-MC), as it is well-established that HCMV infects and replicates in fibroblasts and neuronal cell types [18,19,63].

Figure 2 shows viability of both cell lines as a function of SPGG concentrations reaching up to 100 µM. Even after a 24-h incubation period, both cell lines exhibited essentially no morphological changes from the mock-treatment control (not shown). Likewise, neither cell line displayed any measurable increase in LDH activity even at the highest dose of SPGG. These results suggested that SPGG displayed no measurable cytotoxicity up to 100 µM in the tested cell lines. In fact, the LC_50_ (lethal concentration for 50% cell death) is likely to be several fold higher because of complete absence of increase in LDH activity at 100 µM.

### 2.2. Tracking of Fluorescently-Labeled HCMV Entry in Presence of SPGG

To assess whether SPGG inhibits entry of HCMV into HFF-1 and SK-N-MC cells, we utilized a fluorescently-labeled virus BAD32GFP, which carries a green fluorescent protein (GFP) tag for rapid quantitation and location of the virus. This viral strain is equally potent in cell infectivity [64].

Preliminary studies with BAD32GFP were performed to assess infectivity of HFF-1 and SK-N-MC cells. Following a 2-h incubation at RT at an MOI of 5.0, we observed excellent infectivity of the viral strain for both cell-types as monitored by the presence of multiple GFP punctae (aggregates of virions; Figure 3, panels A and D). Addition of SPGG resulted in reduced number of GFP-puncta for both cell types (Appendix A). Using ImageJ, the immunolabeled punctae with SPGG treatment were quantified as compared to the mock-treated cell lines. The relative decrease in GFP-puncta as a function of SPGG dose followed the traditional semi-log relationship with an *IC*_50_ of 2.0 ± 0.3 µM for HFF-1 and 1.8 ± 0.2 µM for SK-N-MC cells (Figure 3 panels C, and F). In fact, at the maximal dose studied (100 µM), SPGG treatment almost completely abolished BAD32GFP internalization (Figure 3B,E). This implied that SPGG exhibit excellent ‘therapeutic’ window of more than 50 and 55-fold (ratio of LC_50_/IC_50_), respectively for the two cell lines.

### 2.3. The Impact of SPGG Treatment on HCMV Immediate Gene (IE) Expression

*A priori*, an anti-viral inhibition strategy revolving on competitive antagonism to host cell entry does not have to rely on any changes on gene expression, either for the virus or for the host. However, evidence of gene expression changes, especially of those involved in viral entry, propagation and spread, may help better understand an agent’s mechanism of action. Thus, we focused on whether the expression of immediate early (IE) genes of HCMV are altered.

We monitored IE gene expression using AD169, a widely characterized HCMV strain [65], using a cell-ELISA assay (Figure 4). Control mock-treatment (PBS) experiments with HCMV strain AD169 showed robust expression of both IE 1 and 2 genes within one hour of incubation with host cells. Pre-treatment of AD169 strain with SPGG followed by incubation with either HFF-1 or SK-N-MC cells showed significant reductions in expression of both IE1 and IE2. A clear dose-dependent decrease in IE 1 and 2 expression was evident, which gave IC_50_s of 3.1 µM (HFF-1) and 5.6 µM (SK-N-MC) (Figure 4A,B). As with puncta formation, the best results were observed at 100 µM SPGG. Thus, the results show that SPGG-mediated interference in viral entry negatively impacts the expression of viral genes, which probably contribute to reducing the viral infectivity.

Our study is first to report a unique heparin mimetic, which has been previously been reported as a broad spectrum antimicrobial activity, as an inhibitor of HCMV. Earlier, we have demonstrated that SPGG inhibits HSV [50,51] and *Chlamydia* infection [66]. Since HSV and *Chlamydia* are known to exploit heparan sulfate during early stages of host pathogen interactions [66,67], SPGG’s lowering of HCMV early gene expression is not too surprising considering that the viral entry is blocked in the first place. A more important point of this result is the possibility that SPGG could be utilizing multiple mechanisms for its antiviral effects.

### 2.4. The Impact of SPGG Treatment on HCMV Spread

The effect observed on the expression of important viral genes led to the prediction that SPGG possibly does not just function as a heparan sulfate competitor. We reasoned that SPGG may possibly bind to proteins involved in viral spread too. To test this hypothesis at a morphological level, rather than at a molecular level, we studied the phenomenon of viral spread using a plaque reduction assay. In this assay, we used β-galactosidase-expressing reporter HCMV strain (RC256 from ATCC), which upon expression of β-galactosidase and x-gal staining in the infected cells showed blue foci. Plaque reduction assay have been considered the gold standard for antiviral susceptibility testing [68].

The plaque reduction assay was performed on HFF-1 cells using SPGG-treated or PBS mock-treated reporter virus. Initial experiments were performed to deduce the optimal concentration of MOI and period of infection with wild-type virus to detect foci to aid study of cell-to-cell spread. Treatment of HFF-1 with mock-treated HCMV strain RC256 for seven-days followed by x-gal staining and quantification of the blue-colored infected foci under 10× magnification led to highly reproducible measurement of viral spread. When 100 µM SPGG was used to pre-treat the virions, significantly fewer foci were observed after 7 days. Although the punctae observed in the assays may be aggregates of multiple viral particles, counting the number of viral foci showed a dramatic decrease in comparison to the mock-treated HCMV (Figure 5). This is especially important because the effect was measured after 7 days of treatment. Thus, these results at the morphological level indicated that SPGG also inhibited HCMV cell-to-cell spread.

### 2.5. SPGG Inhibition of HCMV Entry Arises Partially from Binding to Glycoprotein B (gB)

As described above, SPGG is a highly sulfated agent that can theoretically bind to several HS-binding proteins. One of these proteins is glycoprotein gB of HCMV, which is a known heparin-binding protein [69,70]. In fact, the design of this work, was based on the expectation that SPGG would bind to one of the viral surface glycoproteins. To test this expectation, we studied whether SPGG binds to gB using dot-blot hybridization and cell-ELISA assay. Figure 6 (panels A,B) shows the results of the dot-blot in the presence and absence of SPGG. As the concentration of SPGG increased to 100 µM, the gB signal reduced 30.5 ± 10% of the untreated control. Although a reduction in signal intensity at lower concentrations (<10 µM) was noticeable, the inhibition was not as robust as observed with IC_50_ profiles of viral entry or expression of genes (Figure 3 and Figure 4 above). To further test whether SPGG engages viral entry glycoprotein gB, we studied Chinese Hamster Ovary cells (CHO-K1) that over-express HCMV gB on cell surface. Dot-blot hybridization and cell-ELISA of CHO-K1 cells in the presence of 100 µM SPGG also led to reduction in gB signal intensity of 34.6 ± 20% in comparison to mock-treated controls (Figure 6C).

Together, two independent experiments with two different types agents bearing glycoprotein gB, i.e., HCMV virus and CHO-K1 cells, exhibited almost equivalent results for 100 µM SPGG. These results are encouraging and likely support that SPGG binds to glycoprotein gB of HCMV, although its affinity is expected to be weaker (10 to 100 µM). One explanation for this is that the anti-gB antibody and SPGG do not compete ideally for the same site of binding on gB. It is possible that both competitors bind gB simultaneously, albeit in a partially competitive manner, resulting in a less robust competition. Another possibility is that SPGG mediates HCMV inhibition through multiple viral surface receptors, one of which is gB. The foundation for this expectation is that SPGG has been earlier shown to bind tightly to glycoprotein D of HSV [50,51]. Likewise, SPGG also inhibits *Chlamydia* infection through binding to its cell surface receptors [66]. Finally, potential pathways that could be targeted by SPGG do exist, e.g., the engagement of one or more host cell surface receptors (e.g., PDGF-R, EGF-R, neuropilin, or others). Of the two, the latter is more likely because heparan sulfate and its mimetics are known to be pleiotropic entities.

## 3. Discussion

This work demonstrates for the first time the concept that small synthetic sulfated agents could effectively inhibit HCMV entry into host cells. Although previous work has demonstrated the concept that certain sulfated, natural, or unnatural polysaccharides (e.g., dextran sulfate, pentosan polysulfate, heparin, copolymers of acrylic acid) can inhibit HCMV infectivity in CHO-K1 and MRC-5 cells [71], the foundation for this activity was based on mimicking the polymeric scaffold of heparan sulfate, which has now been shown to be critical for HCMV entry [72,73,74,75]. In fact, the plausible molecular basis for this competitive inhibition was the interaction of sulfated polymers to viral glycoprotein gB of HCMV [25,26,27]. More specifically, the competitive inhibition was predicted to arise from mimicking the structure of certain heparan sulfates, e.g., 3-*O* sulfated and 6-*O* sulfated species [74]. In stark contrast, SPGG is a much smaller sulfated entity that performs the mimicking function with excellent potency. This is important because a smaller molecular scaffold is easier to transform into clinical drug candidates. Whereas polymers are difficult to synthesize, characterize, purify, analyze, monitor, and administer, smaller molecules are the exact opposite. Thus, SPGG provides the first small molecule lead toward discovery of anti-HCMV drugs.

The 2nd major reason for the novelty behind the discovery of SPGG is the possibility of simultaneous engagement of multiple receptors involved in HCMV entry into host cells. Complex interactions between viral envelope glycoproteins and host cell receptors come into play for successful entry [23,24]. One reason why nature seems to have engineered multiplicity of such interactions is to ensure probability of success. To significantly reduce efficiency of entry, it is important to simultaneously impact these interactions. Thus, targeting only interactions of 3-*O*-sulfated or 6-*O*-sulfated species, inhibition of HCMV entry may not yield a clinically viable agent. Despite being small, SPGG presents a diverse library of sulfated species that could bind all possible heparan sulfate binding receptors, thereby impeding viral recognition. This appears to be the case because SPGG binding to glycoprotein gB explains only part of the inhibition effect. Thus, pleiotropicity of SPGG interactions may be a key reason for its high anti-HCMV potency.

Another reason behind the excitement with SPGG is the inhibition of the immediate early viral gene expression in two different cell lines. As yet no sulfated biopolymer has been demonstrated to display this phenomenon. This finding is important because the products of immediate early genes are essential for viral DNA replication. Therefore, SPGG-like small molecule that display this phenomenon may offer novel strategies for inhibiting HCMV. Likewise, the reduction in viral spread adds more value to the inhibitory role of SPGG. It is important to note that the teratogenicity and toxicity of available HCMV antiviral agents make treatment options during early development markedly limited.

Although SPGG appears to be very valuable in terms of HCMV inhibition, it is important to note that it is still a mixture of septa-sulfated to dodeca-sulfated species. In terms of drug discovery, this is still a considerable limitation and better analogs of SPGG will have to be designed and evaluated. It would also be important to identify the receptor targets engaged by SPGG. Likewise, it would be also important to identify whether certain viral entry pathways are preferentially targeted by SPGG to develop strategies for further drug development. Thus, SPGG is only the first probe for anti-HCMV drug discovery.

In summary, our results provide the first evidence in support of pleiotropicity as a valuable approach for discovering anti-HCMV agents. We project that by simultaneously blocking multiple steps involved in HCMV entry and spread, SPGG holds good promise for future drug development.

## 4. Material and Methods

### 4.1. Cells and Viruses

HFF-1 cells (ATCC SCRC-1041), SK-N-MC cells (ATCC HTB-10), and CHO-K1 cells (ATCC CRL-9618) were purchased from the American Type Culture Collection (ATCC, Manassas, VA, USA) and were used for in vitro studies. Cells were grown under standard culture conditions (37 °C, 5% CO_2_) and were passaged according to manufacturer’s recommendations. HFF-1 and SK-N-MC cells were grown in Dulbecco’s modified Eagle medium (DMEM) and CHO-K1 cells were grown in Hams F12 medium (Corning, Tewksbury, MA, USA) supplemented with 100 U/mg/mL penicillin/streptomycin (Sigma-Aldrich, Saint Louis, MO, USA), and 10% fetal bovine serum (Optima, R&D Systems, Flowery Branch, GA, USA). HCMV strain RC-256 was purchased from ATCC (ATCC VR-2356). This recombinant derivative of the Towne strain carries and expresses the *E. coli* lac*Z* gene. HCMV strain AD169 (ATCC VR-538) was also purchased from ATCC. HCMV strain BAD32GFP (from the Tandon lab), is an AD169 derivative fused with green fluorescent protein (GFP) [76]. Viral stocks were propagated in HFF-1 cells and titers were determined by plaque and immunofluorescence assays and the resulting stocks were stored at −80 °C.

### 4.2. SPGG

The synthesis of SPGG has been reported earlier [77]. Briefly, it involves five steps starting from commercially available raw materials and culminating in a microwave-based sulfation reaction. The overall yield of this five step process is ~40%. The characterization of SPGG and its variants have been described in a later report [61]. The variant of SPGG used in this report has defined molecular composition of septa- (6%), octa- (17%), nona- (21%), deca- (45%), undeca- (11%), and dodeca- (3%) sulfated species (average molecular weight 2178; see Al-Horani et al., 2013), which has been reproducibly synthesized and characterized using advanced LC-MS/MS methods. This SPGG variant is most studied variant so far through characterization of anti-thrombotic, anti-HSV and anti-*Chlamydia* activity profiles [66].

### 4.3. Plaque Reduction Assay

HFF-1 cells were grown to semi-confluency overnight in a 12-well plate. The following day, RC256 was pretreated with 100 µM SPGG in serum free media (SFM) for 1 h at room temperature (RT) with agitation. Cells were infected with treated or untreated virus at a multiplicity of infection (MOI) of 1.0 in SFM for 2 h with rocking. After viral absorption, samples were washed three times in phosphate buffered saline (PBS) before the addition of overlay media (DMEM containing 1% heat-inactivated FBS and 0.5% methyl cellulose). Once a cytopathic effect was observed (four-seven days post infection), overlay media was removed and wells were washed three times with PBS. Cells were then fixed with fixative buffer (4% paraformaldehyde 0.2% glutaraldehyde in PBS) for 30 min, washed three times in PBS and incubated in permeabilization buffer (2 mM MgCl_2_, 0.01% Deoxycholate, 0.02% Nonidet P40 in PBS) for 30 min. Cells were then incubated in a substrate solution containing 500 µg/mL 5-bromo-4-chloro-3-indolyl-β-galactoside (X-gal, Thermo Fisher, Waltham, MA, USA). Blue-stained infected cells were counted under 10× magnification on an inverted microscope.

### 4.4. LDH Cytotoxicity Assay

Cellular toxicity of SPGG (0.39100 µM) was evaluated after 24 h using lactate dehydrogenase (LDH) release as an indicator of damage to the plasma membrane. The assay was performed by using a cytotoxicity assay kit (Thermo Fisher, Waltham, MA, USA) as previously described [50].

### 4.5. Cell-ELISA

HFF-1 or SK-N-MC cells were grown to semi-confluency overnight in black opaque 96-well plates. The following day, AD169 was preincubated with different dilutions of SPGG for 1 h with agitation and cells were infected with 0.3 MOI of treated or untreated virus for 2 h at RT. The infection media was replaced with SFM and incubated for an additional 5 h under standard conditions. The samples were fixed for 20 min with methanol, washed three times in Tris-buffered saline (TBS), and blocked for 2 h with protein-free blocking buffer (Thermo Fisher, Waltham, MA, USA). Samples were incubated overnight at 4 °C with a mouse monoclonal antibody against immediate early genes 1 and 2 (Virusys, Taneytown, MD, USA, cat. no. P1215) diluted 1:5000 in blocking buffer. Next, samples were washed three times with wash buffer (0.05% tween 20 in TBS) and then incubated for 1 h at RT with goat anti-mouse IgG (H+L) peroxidase-conjugated secondary antibody (Thermo Fisher, Waltham, MA, USA) diluted 1:20,000 in wash buffer. Samples were washed 3 times in wash buffer, Super Signal ELISA Femto Substrate (Thermo Fisher, Waltham, MA, USA) was added, and total luminescence was analyzed using a microplate luminometer (Beckman DTX 880).

### 4.6. Confocal Microscopy

HFF-1 and or SK-N-MC cells were seeded on cover glass in a 24-well plate and grown to 40% confluency overnight. Virus strain BAD32GFP was preincubated with dilutions of SPGG for 1 h with agitation. Cells were infected with 5.0 MOI of treated or untreated virus for 2 h at RT. The infection media was replaced with SFM and incubated for an additional 2 h under standard conditions. Samples were washed three times in PBS, fixed for 30 min, and washed an additional three times. Samples were then incubated overnight at 4 °C with anti-GFP mouse IgG2a secondary antibody (Thermo Fisher, Waltham, MA, USA) diluted 1:500 in blocking buffer. Samples were washed three times in PBS and incubated in normal goat serum diluted 1:100 in wash buffer overnight at 4 °C. Samples were then washed 3 times in PBS and then incubated with goat anti-mouse IgG Alexa Fluor 488 (Thermo Fisher, Waltham, MA, USA) diluted 1:500 in wash buffer for 1 h at RT. Samples were washed three times in PBS and incubated in phalloidin (Thermo Fisher, Waltham, MA, USA) diluted 1:40 in blocking buffer for 30 min in a humidity chamber. Samples were washed two times in PBS 0.2% Triton X-100, and once in dH_2_O before mounting to slides with hardset mounting media with DAPI (Vector Laboratories, Burlingame, CA, USA). Slides were imaged using the Nikon A1R confocal microscope and images were processed using ImageJ (version 1.52p, National Institutes of Health, USA) [78] and the GFP-positive punctae were enumerated in 3 random views per slide.

### 4.7. Cell-ELISA on CHO-K1 Cells Expressing HCMV gB

CHO-K1 cells were plated on 96-well plates and grown to 70%-90% confluence. The following day, CMV gB plasmid (pCCMVgB) from the Tandon lab, was transfected into cells using Lipofectamine 2000 (ThermoFisher, Waltham, MA, USA) and incubated overnight. Transfection media was replaced with complete media to allow cell recovery. Following, transfected cells were incubated with dilutions of SPGG for 1 h in SFM under standard conditions. The cells were then washed in TBS, fixed with methanol for five minutes and washed again. Cell-ELISA protocol described previously was then followed using 0.5 µg/mL monoclonal antibody to HCMV gB (Acris, Rockville, MD, USA, cat. no. BM3261) and goat anti-mouse peroxidase conjugated secondary antibody diluted 1:10,000 (ThermoFisher, Waltham, MA, USA). Final washing of the plate was done three times in TBST before 3,3′,5,5′-tetramethylbenzidine (TMB) substrate (Pierce Biotechnology, Rockford, IL, USA) was added and the reaction was stopped. The enzymatic activity was measured at OD 450 nm by microplate photometer (Thermo Scientific Multiskan FC).

### 4.8. Dot Blot Analysis of HCMV for gB in the Presence of SPGG

HCMV strain AD169 was pre-incubated with dilutions of SPGG for 1 h with agitation at RT. Following incubation, 10 µL spots containing 5 × 10^5^ PFU AD169 and SPGG were applied to a 0.2 µm nitrocellulose membrane (Bio-Rad Laboratories, Hercules, CA). Spots were dried and the membrane was blocked in protein-free blocking buffer (ThermoFisher, Waltham, MA, USA) for 1 h. Standard ELISA protocol previously described was then followed using 1 µg/mL monoclonal antibody to HCMV gB (Acris) and goat anti-mouse peroxidase conjugated secondary antibody diluted 1:10,000 (ThermoFisher). After the final wash, the blot was incubated with 3-amino-9-ethylcarbazole (AEC) staining solution (Sigma-Aldrich, St. Louis, MO, USA) for 30 min and rinsed in water before imaging. The intensity of the signal was quantified using ImageJ gel analysis function to calculate and plot the area under each peak.

### 4.9. Statistical and Software Analysis

All experiments were performed in triplicate unless otherwise stated to confirm consistency of results. The student t-test and ANOVA was used to analyze data and a *P*-value of less than 0.05 was considered statistically significant between the control and the treatment groups. ImageJ was used to enumerate GFP signal in random sections of infected cells for confocal microscopy and was also used to estimate relative density and size of samples in the dot blot.

## Figures and Tables

**Figure 1 ijms-21-01676-f001:**
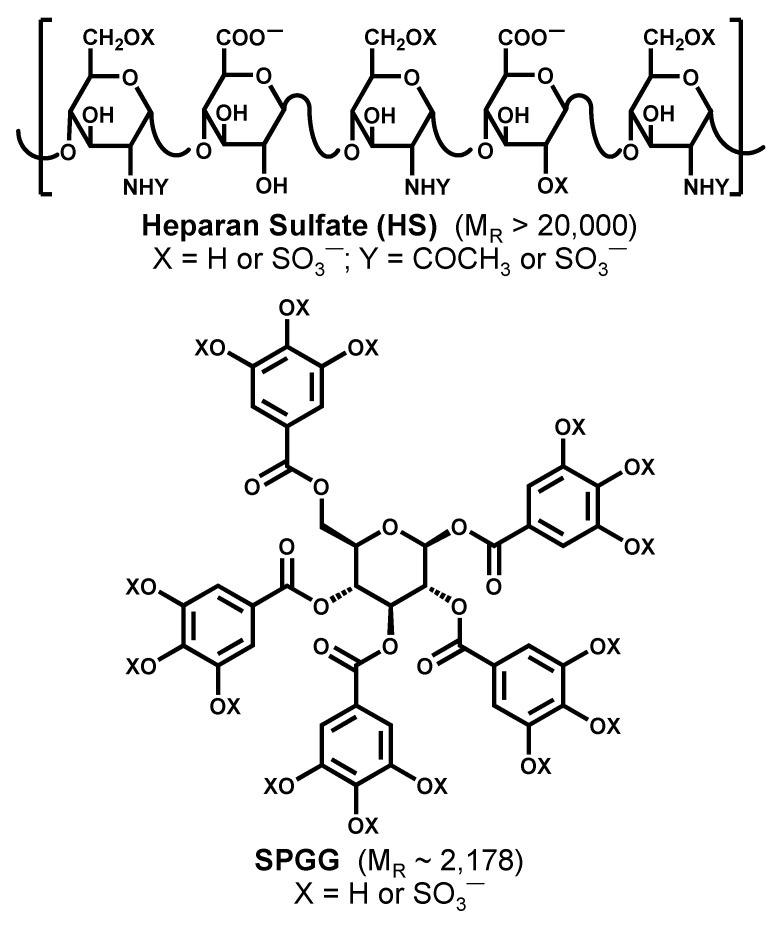
Heparan sulfate (HS) and sulfated pentagalloylglucoside (SPGG). Whereas HS is a linear polymer with an average molecular weight (Mr) of approximately 20,000, SPGG is a globular small molecule with Mr of 2178. HS is a biopolymer; in contrast SPGG is synthesized in 5 steps in excellent yields.

**Figure 2 ijms-21-01676-f002:**
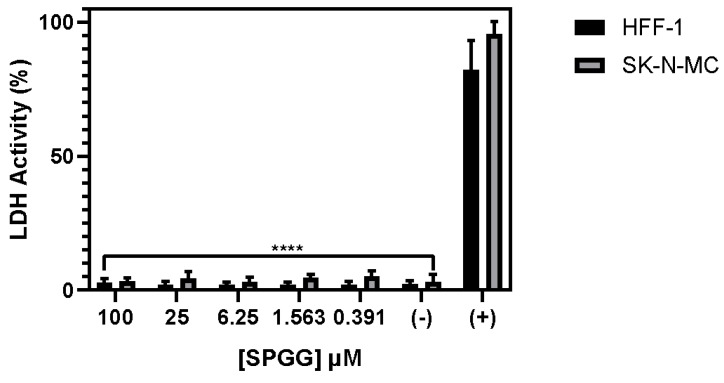
Effect of SPGG on HFF-1 and SK-N-MC cells. The cells monolayers were incubated in the presence or absence of SPGG in a dose dependent manner. Controls; (+) represents maximum lactate dehydrogenase (LDH) activity control, (−) represents the spontaneous LDH release control. A one-way analysis of variance (ANOVA) determined significant difference among means with a *P*-value of < 0.0001 in both cell types. Dunnett’s multiple comparison test found significant differences between the mean of each treatment in comparison to the mean of the positive control column, (****) signifies a *P*-value of < 0.0001. The graph is indicative of the mean values and SD of triplicate experiment.

**Figure 3 ijms-21-01676-f003:**
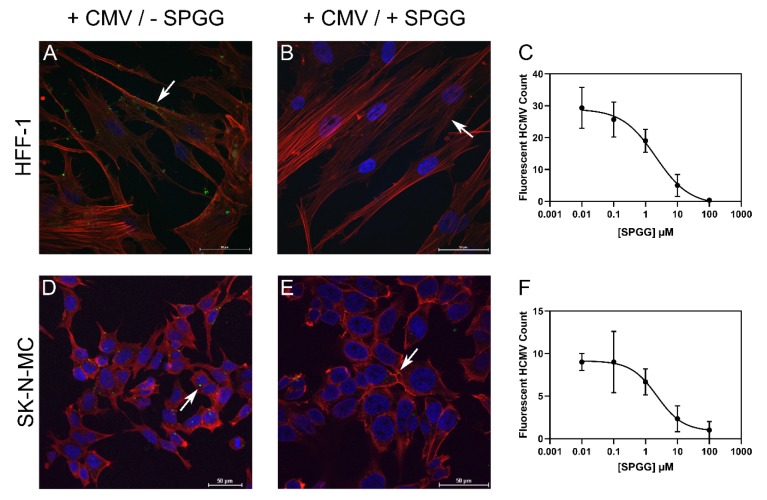
Human cytomegalovirus (HCMV) entry into HFF-1 and SK-N-MC cells treated with SPGG in High-resolution confocal microscopy. HFF-1 and SK-N-MC cells were challenged with the HCMV strain BAD32GFP which was pre-incubated with or without SPGG 100µM for 1 h. The double immunolabeling in confocal microscopy shows an increased number of GFP puncta (green highlighted with white arrows) in HFF-1 cells (**A**) and SK-N-MC cell (**D**) infected with HCMV and mock-treated compared to HCMV treated with SPGG (**B**,**E**). Cells were counterstained with phalloidin (red) and DAPI (blue) and images were processed and enumerated using ImageJ. The number of GFP-positive puncta detected in HFF-1 (**C**) and SK-N-MC (**F**) cells infected with HCMV exposed to different concentrations of SPGG was quantified in ImageJ using 3 random views per slide. The Graphs show a clear dose-dependent reduction in number of GFP-positive for both cell lines. Data are shown as mean value and SD for a triplicate experiment. Scale bar (**A**–**E**) = 50 µm.

**Figure 4 ijms-21-01676-f004:**
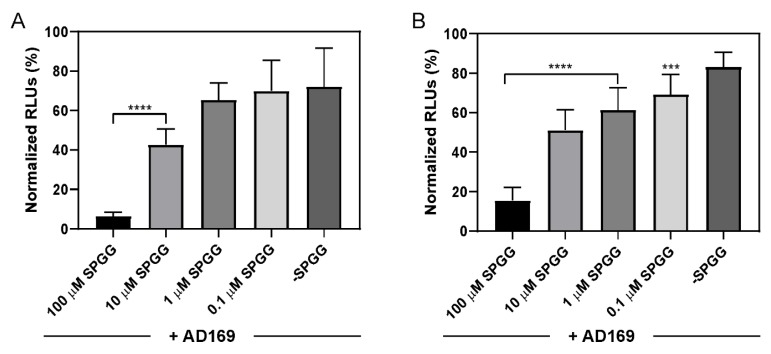
HCMV immediate early (IE) gene 1 and 2 expression in (**A**) HFF-1 cells and (**B**) SK-N-MC cells in the presence of SPGG. In this cell-ELISA experiment, HFF-1 (**A**) or SK-N-MC (**B**) cells were challenged with HCMV (0.3 multiplicity of infection (MOI)) pre-treated with SPGG or mock and an antibody against immediate early (IE) genes. The graphs reveal a SPGG-dose-dependent decreased expression of IE both in HFF-1 and SK-N-MC cell line, respectively. ANOVA determined significant differences among means with a *P*-value of < 0.0001 in both (**A**) and (**B**). Further analysis using Dunnett’s multiple comparison test found significance of decreased expression when compared to the mean of the control column, (****) signifies a *P*-value of < 0.0001 and (***) signifies a *P*-value of 0.0002. The graphs result from mean values and SDs of experiments, each condition was tested in triplicate at N of 5.

**Figure 5 ijms-21-01676-f005:**
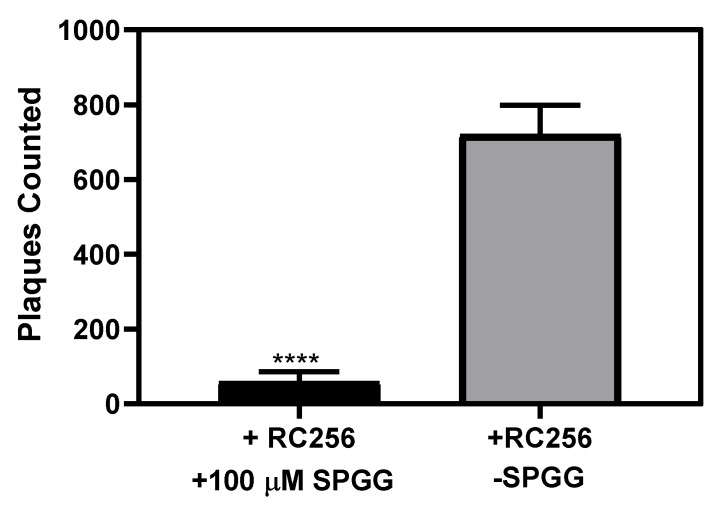
HCMV-mediated foci formation using plaque reduction assay in presence of SPGG. In the experiment, HFF-1 cells were infected with an MOI of 1.0 of HCMV β-galactosidase-expressing reporter virus strain (RC256) pretreated with 100 µM SPGG or mock-treated in serum free media (SFM). The number of plaques was quantified in both treated and untreated samples and revealed a dramatic decrease in samples pretreated with 100 µM SPGG compared to mock treatment. The graphs are the result of mean values and SD for a N = 3 experiments with triplicates of each conditions. Statistical significance was determined with a T-Test, (****) signifies a *P*-value of < 0.0001.

**Figure 6 ijms-21-01676-f006:**
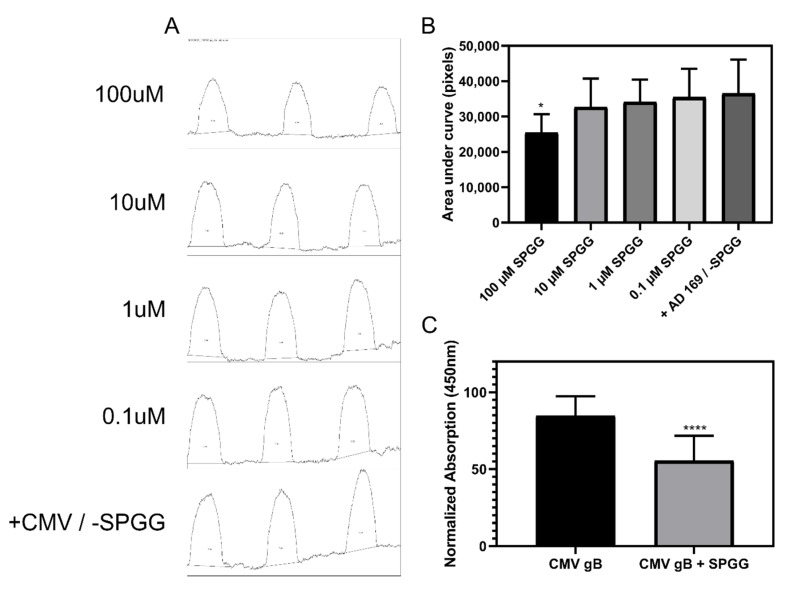
Displays moderate affinity for HCMV entry glycoprotein B (gB) in two independent assays. Panels (**A**) and (**B**): Dot blot analysis showing one possible mechanism behind SPGG’s inhibition of HCMV entry. Spots containing 5 × 10^5^ PFU of HCMV AD169 strain and dilutions of SPGG were applied to nitrocellulose membrane and an immunoassay was carried out. The signal was then quantified for size and pixel density using ImageJ gel analysis function and the profile plot was generated with each curve representing a respective spot on the original blot (panel **A**). The area under each curve in (**A**) was then calculated and plotted in (panel **B**). Panel (**C**): Cell-ELISA was used to assess the affinity of SPGG for HCMV gB over-expressed in CHO-K1 cells. Each experimental condition was tested in at least triplicate with an N = 4. The errors in (**B**) and (**C**) correspond to + SD. ANOVA determined significant differences among means with a *P*-value of 0.0273 (*) in (**B**). Further analysis of (**B**) using Dunnett’s multiple comparison test found a significant difference in the 100 µM treatment when compared to the mean of the positive control with a *P*-value of 0.0117 (*). T-test analysis was used to determine significance in (**C**). (****) signifies a *P*-value of < 0.0001.

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
