# Peer review of "Inhibition of Human Cytomegalovirus Entry into Host Cells through A Pleiotropic Small Molecule"

_ijms, 2020, doi:10.3390/ijms21051676_

Round 1
Reviewer 1 Report
In this report, Elste and colleagues demonstrated the antiviral effects of a synthetic mimetic of heparan sulfate, sulfated pentagalloylglucoside (SPGG) on human cytomegalovirus (HCMV) in HFF-1, SK-N-MC and CHO-K1 cells. They used confocal microscopy, cell-ELISA and plaque reduction assay to show that SPGG inhibits HCMV entry and spread during infection of cells. Additionally, they showed that SPGG downregulated the expression of early gene 1 and 2. This work is important because they suggested that SPGG is capable of simultaneously targeting multiple receptors of HCMV and thus a good candidate for more effective therapy.
Comments:
The results need editing so that data are presented more clearly. For example, the first two paragraphs of the results should be moved to introduction. Lines 265 - 266: "This...... Mississippi)" should be removed. The statement should remain in methods Lines 273 - 274: "which .... yield" should be replaced with the word "with" Figures 3A and D should include arrows pointing to the green spots since they are difficult to see. Figures 3C and D should include units of measurement. Was the count done per view or per slide? Line 292 is not clear. Please clarify with the missing word Include Figure 4A and 4B specifically in the text probably in line 300 Line 310: Please remove one of the repeated HFF-1 Lines 312 - 313: Please edit the statement "SPGG .... SK-N-MC" for clarity. Figure 5 shows the number of plaques and nothing about plaque size unlike what was written in the text. What is the unit of measurement? Were the plaques counted per well or per area? Figure 5 label should also include "+RC256" in the presence of SPGG Line 348: "To gain test" should be corrected Line 351: 30.5% and line 356: 34.6% should include standard deviation (+/-) The error bars shown on Figures 6B and C overlap and do not support the statistical significance reported. It appeared that the authors did not use the correct statistical analysis. The authors need to use the non-parametric Mann-Whitney U test for the analysis of the data. They should thus use the new statistical results to explain section 3.6 and adjust their conclusions accordingly. Line 402: I suggest the authors remove the words "In fact" from the statement.
Author Response
Reviewer 1
“In this report, Elste and colleagues demonstrated the antiviral effects of a synthetic mimetic of heparan sulfate, sulfated pentagalloylglucoside (SPGG) on human cytomegalovirus (HCMV) in HFF-1, SK-N-MC and CHO-K1 cells. They used confocal microscopy, cell-ELISA and plaque reduction assay to show that SPGG inhibits HCMV entry and spread during infection of cells. Additionally, they showed that SPGG downregulated the expression of early gene 1 and 2. This work is important because they suggested that SPGG is capable of simultaneously targeting multiple receptors of HCMV and thus a good candidate for more effective therapy.”
Response: We appreciate the reviewer’s comments commending the manuscript.
Comments:
“The results need editing so that data are presented more clearly. For example, the first two paragraphs of the results should be moved to introduction.
Response: We have moved section 3.1 to introduction as suggested.
Lines 265 - 266: "This...... Mississippi)" should be removed. The statement should remain in methods.
Response: Done.
Lines 273 - 274: "which .... yield" should be replaced with the word "with"
Response: Done.
Figures 3A and D should include arrows pointing to the green spots since they are difficult to see.
Response: Done.
Figures 3C and D should include units of measurement. Was the count done per view or per slide?
Response: The counts were done per 3 random views.
Line 292 is not clear. Please clarify with the missing word.
Response: Done.
Include Figure 4A and 4B specifically in the text probably in line 300.
Response: Done.
Line 310: Please remove one of the repeated HFF-1
Response: Done.
Lines 312 - 313: Please edit the statement "SPGG .... SK-N-MC" for clarity.
Response: Done.
Figure 5 shows the number of plaques and nothing about plaque size unlike what was written in the text. What is the unit of measurement? Were the plaques counted per well or per area?
Response: The number of plaques were measured. We have revised the text to reflect this point.
Figure 5 label should also include "+RC256" in the presence of SPGG
Response: Done.
Line 348: "To gain test" should be corrected
Response: Done.
Line 351: 30.5% and line 356: 34.6% should include standard deviation (+/-)
Response: We have added this information to the text.
The error bars shown on Figures 6B and C overlap and do not support the statistical significance reported. It appeared that the authors did not use the correct statistical analysis. The authors need to use the non-parametric Mann-Whitney U test for the analysis of the data. They should thus use the new statistical results to explain section 3.6 and adjust their conclusions accordingly.
Response: In the revised manuscript, we have reported description of the statistical test used to analyze the data and also included the revised statistical significance.
Line 402: I suggest the authors remove the words "In fact"
Response: Revised.
Reviewer 2 Report
Elste et al in the manuscript “Inhibition of Human Cytomegalovirus Entry into Host Cells Through a Pleiotropic Small Molecule” reported that SPGG, a functional mimetic of heparan sulfate, was able to block HCMV infection by targeting multiple entry factors/receptors. The manuscript is overall well written, and the results seem to be rather promising with minimal cytotoxicity even at 100 µM. However, there exist a few cases of evidence that are unable to justify the conclusion as stated in the text, e.g., the effects of SPGG on HCMV IE gene expression and on HCMV spread. I am afraid the manuscript at the current stage needs to be intensively improved to be published elsewhere.
The authors stated that the SPGG negatively impacted the HCMV IE expression (session 3.4, page 8, line 290-308). However, how the authors rule out the effects of SPGG on viral entry? It appears to me that the reduced IE gene expression was caused by inhibition of HCMV entry since the IC50 of entry inhibition curve (Fig 3C and 3D) is so close to IC50 of IE gene inhibition curve (Fig 4A and 4B). Similarly, the HCMV spreading attenuation (Session 3.5, page 9, Line 318-335) may simply be a consequence of compromised viral entry. One way to evaluate the viral spreading is not counting the number of plaques but instead quantifying the plaque size. It needs to be cautious to interpret the results by using GFP tagged CMV. The puncta shown in Fig 3A, B, D, E may not be the real viral particles but instead the aggregates of virions. Since the size of the single viral particles (generally <200nm), the scales (50 µM) shown under the confocal microscope unlikely reveal a single HCMV particle. The assays “Cell-ELISA on CHO-K1 cells expressing HCMV gB” and “Dot Blot assay” are confusing. Why the anti-gB antibody can differentiate the gB proteins bound or unbound to SPGG? Because the anti-gB share the same epitope as the SPGG? Any evidence? The authors claim that the SPGG targets multiple entry receptors and thus confers greater inhibitory potency, but evidence/reference is required to prove that.
Author Response
Reviewer 2
“Elste et al in the manuscript “Inhibition of Human Cytomegalovirus Entry into Host Cells Through a Pleiotropic Small Molecule” reported that SPGG, a functional mimetic of heparan sulfate, was able to block HCMV infection by targeting multiple entry factors/receptors. The manuscript is overall well written, and the results seem to be rather promising with minimal cytotoxicity even at 100 µM.”
Response: We appreciate the reviewer’s comments commending the manuscript.
“However, there exist a few cases of evidence that are unable to justify the conclusion as stated in the text, e.g., the effects of SPGG on HCMV IE gene expression and on HCMV spread. I am afraid the manuscript at the current stage needs to be intensively improved to be published elsewhere.”
Response: We appreciate the reviewer’s critique and address these comments in the revised manuscript. See below for details.
“The authors stated that the SPGG negatively impacted the HCMV IE expression (session 3.4, page 8, line 290-308). However, how the authors rule out the effects of SPGG on viral entry? It appears to me that the reduced IE gene expression was caused by inhibition of HCMV entry since the IC50 of entry inhibition curve (Fig 3C and 3D) is so close to IC50 of IE gene inhibition curve (Fig 4A and 4B). Similarly, the HCMV spreading attenuation (Session 3.5, page 9, Line 318-335) may simply be a consequence of compromised viral entry.”
Response: The reviewer is correct in pointing out that HCMV IE expression could be because of viral entry inhibition. It is possible that our description of results led reviewer to conclude that SPGG was being described as an agent that decreases IE expression without an effect arising from compromised viral entry. In fact, our manuscript first describes the compromised viral entry results and then shows subsequent effects on IE expression. We meant to convey that the overall effect of decrease viral entry were reflected in the subsequent steps in HCMV life cycle, as evidenced by a significant decrease IE gene expression and decease in viral foci formation. We have revised the text to clarify this point.
“One way to evaluate the viral spreading is not counting the number of plaques but instead quantifying the plaque size. It needs to be cautious to interpret the results by using GFP tagged CMV. The puncta shown in Fig 3A, B, D, E may not be the real viral particles but instead the aggregates of virions. Since the size of the single viral particles (generally <200nm), the scales (50 µM) shown under the confocal microscope unlikely reveal a single HCMV particle.”
Response: We appreciate the reviewer’s point about an alternative way to assess virus spread. However, the plaque reduction assay has been considered the gold standard for antiviral susceptibility testing (Wentworth and French, 1970) and hence we used this gold standard in our work. We also agree with the reviewer that the puncta may not be single virions. However, even here it is very significant that following 7 days of infection with HCMV, the presence of SPGG greatly reduced the number of foci formation, which is a measure of the virus spread. We have revised the manuscript to alert the reader on this point.
“The assays “Cell-ELISA on CHO-K1 cells expressing HCMV gB” and “Dot Blot assay” are confusing. Why the anti-gB antibody can differentiate the gB proteins bound or unbound to SPGG? Because the anti-gB share the same epitope as the SPGG? Any evidence?”
Response: The reviewer is correct in pointing out that if SPGG and anti-gB antibody may not be competing fully, or ideally, to yield a robust competitive inhibition profile. This could arise from the possibility that the anti-gB antibody and SPGG do not bind to the same site on gB. Yet, the results show some competitive effect, which implies partially competitive profile. A more likely possibility is that SPGG mediates HCMV inhibition through multiple viral surface receptors, one of which is gB. The foundation for this expectation is that SPGG has been earlier shown to bind tightly to glycoprotein D of HSV (Gangji et al., 2018; Majmudar et al., 2019). Likewise, SPGG also inhibits Chlamydia infection through binding to its cell surface receptors (Gallegos et al., 2019). Finally, potential pathways that could be targeted by SPGG do exist, e.g., the engagement of one or more host cell surface receptors (e.g., PDGF-R, EGF-R, neuropilin, or others). We meant to convey this idea in the paper but apparently failed to clarify it well enough. We have revised the manuscript to enhance clarity of this concept. We thank the reviewer.
“The authors claim that the SPGG targets multiple entry receptors and thus confers greater inhibitory potency, but evidence/reference is required to prove that.”
Response: We agree with the reviewer that evidence is required to state that SPGG targets multiple entry receptors. In this work, we have shown that SPGG binds to cell surface gB, which is one possible mechanism of reduction of HCMV entry. However, we have also shown that the gB binding potency does not exactly correlate with the IC50 of inhibition of viral entry. This implied that others targets of SPGG do exist. In fact, we have earlier shown glycoprotein D of HSV and surface protein of chlamydia are the targets of SPGG (Gangji et al., 2018 ACS Med Chem Lett; Majmudar et al., 2019 Antiviral Research; Gallegos et al., 2019 Frontiers Microbiology). This led us to suggest that other targets of SPGG on HCMV surface are possible, and more than likely. Yet, we agree with the reviewer that detailed evidence was not provided. Hence, we have revised our statements for the reasons stated above.
Reviewer 3 Report
In this manuscript, Elste et al. describe a heparan sulfate mimetic that can block entry of cytomegalovirus in cells. They show that preincubation of the virus with this SPGG reduces entry of CMV into cells and expression of early lytic genes and they claim this is not just due to binding to the CMV fusogen gB. Given that there are few drugs against CMV infection and none safe enough to use in pregnancy against congenital CMV, this is an interesting starting point for a drug targeting CMV entry. Nonetheless, I have concerns about their assay choices and some interpretation of data from a virology standpoint.
Major
1) All the assays were done by pre-incubating CMV with SPGG for 1 hr. This clearly is not how SPGG would be used as a drug, although it is a nice starting point to show the potential for effects and . Nonetheless, it would be good if they could show at least one panel were they add SPGG with CMV in the culture medium to show that it can block the virus without the direct preincubation. If not they should temper what they say regarding the potential for CMV entry inhibition, because there is no evidence that SPGG would even work in solution, let alone in tissue.
2) In Figure 4 they show a reduction in early gene expression with SPGG incubation and they seem to suggest this is an interesting result not shown before with heparan sulfate mimetics. While I am not necessarily arguing with the result (although cell ELISA would probably not be the method of choice in a virology lab), to me this is the expected result based on Figure 3. If the virus get in the cells less efficiently, there should be a reduction in viral gene expression. If there was not SPGG would not be useful as a drug. Therefore while the statement in lines 305-307 may be technically true ('Our study is the first ... alters viral gene expression'), it means the other drugs did not actually sufficiently block entry, not that SPGG necessarily have an additional function worth noting. If they want to test for a separate function, they would have to first infect cells, then treat the cells (not the virus) with SPGG and see if there is a reduction in gene expression. They should rephrase this entire section and describe Figure 4 just as additional evidence for the result shown in Figure 3, which is nonetheless an important and valuable point to show the potential for SPGG to work.
3) I am not sure Fig. 6 conclusively shows that other SPGG needs to bind other proteins to work. What is the antibodies against gB bind at a different location than SPGG? Would that not mean SPGG and the antibody could bind at the same time? Could they obtain virions without gB and see if SPGG binds virions +/- gB with the same efficiency? (unfortunately without gB you get no fusion, so I am not sure how you would test whether gB is required for the effect of SPGG functionally).
Minor
1) The whole section 3.1 read to me like it should be part of the introduction, especially as reading the introduction SPGG seemed like it came out of nowhere.
2) t-tests are only appropriate for experiments/panels with two conditions only (like Fig 5, 6C). For panels with >2 bars, and thus >1 tested comparison (Fig. 4, 6B), the authors need to use ANOVA followed by a post hoc corrected pairwise test (probably Dunnett's, since they compare all other bars to a single control).
3) Fig. 3 and S1 would benefit from some arrows to point at the dots - upon printing they were very hard to see.
Author Response
Reviewer 3
“In this manuscript, Elste et al. describe a heparan sulfate mimetic that can block entry of cytomegalovirus in cells. They show that preincubation of the virus with this SPGG reduces entry of CMV into cells and expression of early lytic genes and they claim this is not just due to binding to the CMV fusogen gB. Given that there are few drugs against CMV infection and none safe enough to use in pregnancy against congenital CMV, this is an interesting starting point for a drug targeting CMV entry. Nonetheless, I have concerns about their assay choices and some interpretation of data from a virology standpoint.”
Response: We appreciate the reviewer’s positive comments on the value of work. We address his/her critique below.
Major
“1) All the assays were done by pre-incubating CMV with SPGG for 1 hr. This clearly is not how SPGG would be used as a drug, although it is a nice starting point to show the potential for effects and nonetheless, it would be good if they could show at least one panel were they add SPGG with CMV in the culture medium to show that it can block the virus without the direct preincubation. If not they should temper what they say regarding the potential for CMV entry inhibition, because there is no evidence that SPGG would even work in solution, let alone in tissue.”
Response: The reviewer is correct is suggesting that we should not pre-incubate HCMV. Two points in this regard would hopefully convince the reviewer that pre-incubating cells or pre-incubating the virus would give essentially similar results. 1) SPGG is a highly sulfated, small molecule that rapidly diffuses in solution. Because of its chemical nature, it is not metabolized by cells and maintains its protein binding properties for several days. Thus, cellular clearance mechanisms would not impact its recognition of viral cell surface proteins. It is possible that additional SPGG binding proteins on target cells compete with viral surface glycoproteins for SPGG and decrease its IC50; however, this competition could be overcome through the use of higher dose. 2) In fact, our follow-on experiments show that SPGG is highly efficacious in tissue studies. Using porcine-derived corneal ex-vivo explant model, we found that SPGG reduced reporter virus gene expression (x-gal preliminary results; reported in by Elste et al., 2018 A Heparan Mimetic analog SPGG is a potent inhibitor against herpes simplex virus (HSV) infection. 31st Annual ASPET Meeting-Loyola University Stritch School of Medicine in Maywood, IL (June 22nd 2018). Following further work in vivo, we will be publishing the value of SPGG in preventing infections, especially in the eye. Thus, the idea that SPGG is not likely to work in tissues is incorrect. We have revised the description of these experiments to include the comment made by the reviewer.
”2) In Figure 4 they show a reduction in early gene expression with SPGG incubation and they seem to suggest this is an interesting result not shown before with heparan sulfate mimetics. While I am not necessarily arguing with the result (although cell ELISA would probably not be the method of choice in a virology lab), to me this is the expected result based on Figure 3. If the virus get in the cells less efficiently, there should be a reduction in viral gene expression. If there was not SPGG would not be useful as a drug. Therefore, while the statement in lines 305-307 may be technically true ('Our study is the first ... alters viral gene expression'), it means the other drugs did not actually sufficiently block entry, not that SPGG necessarily have an additional function worth noting. If they want to test for a separate function, they would have to first infect cells, then treat the cells (not the virus) with SPGG and see if there is a reduction in gene expression. They should rephrase this entire section and describe Figure 4 just as additional evidence for the result shown in Figure 3, which is nonetheless an important and valuable point to show the potential for SPGG to work.”
Response: We agree with the reviewer and have revised the section such that Figure 4 is presented as additional evidence for the point made in Figure 3.
“3) I am not sure Fig. 6 conclusively shows that other SPGG needs to bind other proteins to work. What is the antibodies against gB bind at a different location than SPGG? Would that not mean SPGG and the antibody could bind at the same time? Could they obtain virions without gB and see if SPGG binds virions +/- gB with the same efficiency? (unfortunately without gB you get no fusion, so I am not sure how you would test whether gB is required for the effect of SPGG functionally).”
Response: The reviewer is correct in suggesting a model that SPGG and the anti-gB antibody may bind gB simultaneously and thereby not generate a binding profile equivalent to the viral entry inhibition profile. Unfortunately, the experiments are challenging to perform as the reviewer also suggests. Hence, we simply meant to alert the reader that the IC50 profiles are different and likely to mean that more than one targets are engaged. In fact, we have previously shown that glycoprotein D of HSV and surface protein of chlamydia are the targets of SPGG (Gangji et al., 2018 ACS Med Chem Lett; Majmudar et al., 2019 Antiviral Research; Gallegos et al., 2019 Frontiers Microbiology). This led us to make the suggestion that SPGG may be working through binding multiple HCMV surface proteins. Yet in response to the reviewer suggestion, we have revised our description with the above discussion.
Minor
“1) The whole section 3.1 read to me like it should be part of the introduction, especially as reading the introduction SPGG seemed like it came out of nowhere.”
Response: We have moved the section to introduction.
“2) t-tests are only appropriate for experiments/panels with two conditions only (like Fig 5, 6C). For panels with >2 bars, and thus >1 tested comparison (Fig. 4, 6B), the authors need to use ANOVA followed by a post hoc corrected pairwise test (probably Dunnett's, since they compare all other bars to a single control).”
Response: We report the analysis in the revised manuscript using ANOVA.
“3) Fig. 3 and S1 would benefit from some arrows to point at the dots - upon printing they were very hard to see.”
Response: We have added arrows as suggested.
Round 2
Reviewer 1 Report
Elste and colleagues have satisfactorily addressed my concerns about the manuscript. They implied that SPGG can be an extremely effective anti HCMV due to its ability to simultaneously taget multiple HCMV receptors.
Reviewer 2 Report
Thanks for the clarification and I am satisfied with the current version.